# Impact of Voluntary, Community and Social Enterprise (VCSE) Organisations Working with Underserved Communities with Type 2 Diabetes Mellitus in England

**DOI:** 10.3390/healthcare11182499

**Published:** 2023-09-08

**Authors:** Lucie Nield, Sadiq Bhanbhro, Helen Steers, Anna Young, Sally Fowler Davis

**Affiliations:** 1Advanced Wellbeing Research Centre, Sheffield Hallam University, Sheffield S10 2BP, UK; 2Department of Nursing and Midwifery, Sheffield Hallam University, Sheffield S1 1WB, UK; s.bhanbhro@shu.ac.uk; 3Voluntary Action Sheffield, Sheffield S1 4FW, UK; h.steers@vas.org.uk; 4Sheffield Hallam University Health Centre, Sheffield S1 1WB, UK; a.young@nhs.net; 5School of Allied Health, Anglia Ruskin University, Cambridge CB1 1PT, UK; sally.fowler-davis@aru.ac.uk

**Keywords:** voluntary sector, diabetes, VCSE, complex evaluation, self-management, England

## Abstract

The Voluntary, Community and Social Enterprise (VCSE) sector offers services and leadership within the health and care system in England and has a specialist role in working with underserved, deprived communities. This evaluation aims to identify best practices in self-management support for those living with type 2 diabetes mellitus (T2DM) and to develop a theory of change (TofC) through understanding the impact of VCSE organisations on diabetes management. An appreciative inquiry (AI) was carried out and co-delivered using qualitative interviews and an embedded analysis with VCSE partners. A voluntary service coordinated seven VCSE organisations who assisted with recruiting their service users and undertook interviews to identify the impact of existing activities and programmes. People living with T2DM were interviewed about services. Themes were as follows: (a) individual and group activities; (b) trusted services and relationships across the community; (c) long-term engagement; (d) sociocultural context of diet and nutritional choices; (e) experience of adaptation; and (f) culturally appropriate advice and independent VCSE organisations. The structured educational approach (DESMOND) for T2DM was accessed variably, despite these services being recommended by NICE guidelines as a standard intervention. The VCSE offered continuity and culturally appropriate services to more marginalised groups. This evaluation highlights the importance of targeted engagement with underserved communities, particularly where primary care services are more limited. The TofC is a unique insight into the impact of VCSE services, offering bespoke support to manage T2DM, suggesting areas for improvements in capacity and offering the capability to sustain the VCSE sector as an essential element of the T2DM care pathway in England.

## 1. Introduction

In England, the Voluntary, Community and Social Enterprise (VCSE) sector comprises organisations that are independent of the government and are constitutionally self-governing. They exist to represent and advocate for the local community or groups with specific needs and aim to promote social, economic, environmental or cultural benefits [1]. The diversity of these organisations and the delivery of their activities are being recognised for their contribution to the wellbeing of populations within communities [2], and the recent formation of Integrated Care Systems (ICSs) across England aims to address further transdisciplinary working and joined up, coordinated care pathways for individuals [3] This is different from the international non-government organisation (NGO) context due to their continuous, bespoke support of VCSEs of marginalised groups in their local communities. The National Institute for Clinical Excellence (NICE) recommend that a variety of self-management support is needed for people with type two diabetes mellitus (T2DM), particularly for those who do not access the typical pathways of care following diagnosis [4]. The academic literature on the VCSE’s role and its impact on health and social care is still being explored but emerging. A joint review (2016) on the partnerships of the VCSE sector in health and social care highlighted that community-based small, local charities; social enterprises; faith-based organisations; and health and social care professional networks often deliver important services, including social groups, community exercise groups and T2DM management services [5]. People living in marginal and underserved communities are sometimes deemed ‘seldom heard’ by statutory services and require the support of VCSE service providers to engage with and meet those communities’ needs [6]. Most recently, COVID-19 has shown how health and care systems and the VCSE sector can be partnered effectively in identifying and developing co-production interventions for specific populations [7].

VCSE organisations play a key role in facilitating dialogue between a system and its residents, ensuring that services are co-produced with residents at the heart of service provision [5]. However, there is a diversity of provision and in many cases best practices are not shared across the many different groups that participate in the planning and provision of community services, particularly where they are contracted to support self-management of specific medical conditions. The wider perceived benefits to VCSE users have been well documented, but further clarity is needed to demonstrate the impact of services [8]. VCSE organisations can play a hugely important coordinating role, brokering between beneficiaries and other stakeholders in the system (e.g., clinical professionals and local authorities). A partnership approach for conditions like T2DM is advantageous for planning sustainable, long-term management [4]. 

Management of T2DM can be complex and requires, among other components, continuous self-management, multidisciplinary team input and community support [9]. The prevalence of T2DM is rising rapidly, putting increased pressure on the statutory healthcare system and challenging the provision of optimal, tailored diabetes care. In England, healthcare is typically accessed via free-of-charge primary care services for individuals experiencing symptoms or concerns regarding their health. Following assessment, people with T2DM receive a medical diagnosis triggering a referral and ongoing monitoring depending on the severity of symptoms and co-morbidities. Person-centred self-management of T2DM is widely recognised as key to achieving better health outcomes [10]. There is a complex range of interacting factors that follow the diagnosis and lead to variation in the quality of long-term management, including emotional and physical challenges [11], relationships with healthcare professionals [12], glycaemic control [12] and psychometric assessments [13]. People with T2DM require tailored, consistent and ongoing information, which the healthcare professional delivers briefly [14], with ongoing referral to the UK’s current structured education programme for T2DM: Diabetes Education and Self-Management for Ongoing and Newly Diagnosed (DESMOND). DESMOND is one of several structured education programmes available in the UK [15] which has been found to improve some clinical outcomes and reduce costs [16]. Its purpose is to increase activation for necessary behaviour change, but this tends to affect only those who are already highly active [17]. It is also more effective in white Europeans [18], despite ethnically diverse individuals being disproportionately represented in UK-based T2DM populations. 

Psychologically, a diagnosis of T2DM can also lead to a ‘biographical disruption’ [19], a term used to identify the reaction to managing a long-term condition. People react variously to the advice offered for weight loss or smoking cessation associated with improving health and can also recognise a stigma with T2DM, often associated with unhealthy lifestyle factors enshrined in household norms and behaviours. The diagnosis can also sit uncomfortably with individuals who viewed themselves as previously healthy. When healthcare professionals focus on the aetiology, diagnosis and treatment of T2DM, the patient may be more concerned with the consequences of diagnosis and the influence on their daily life [14,20]. The perception of the disease process is contextual to social and cultural experience and whilst emotional support may be helpful [21], any intervention must be adapted and endorsed by the patient and ‘fit’ their lived situation [22].

Previous studies in the UK identified a greater risk of diabetes in ethnic minority groups, particularly South Asians [23,24,25]. Several studies have also identified inequalities in access to [26] or compliance and uptake of diabetes services [27,28,29], although not exclusively in ethnically diverse groups. Inequalities exist in primary diagnosis, access to primary healthcare and initial assessment and treatment [30]. T2DM is characterised by an emphasis on individuals making decisions to change their lifestyle, eating habits and exercise regimes; however, some people from poorer communities have difficulty managing personal changes about health based on limited access to support and community assets that enable healthier patterns [31]. 

This research aimed to critically engage with and understand the role of VCSE organisations and their impact on the self-management of people with T2DM from underserved communities and to produce a theory of change (TofC) [32]. A TofC is defined as a theory of how and why an initiative works [33] and is increasingly used in evaluations to capture, as an outcome of qualitative and mixed methods analyses, the initial programme and how it aims to improve the outcomes for a specific population. A TofC describes an impact pathway, seeking to demonstrate the causal assumptions behind the links in the pathway. The aim of a TofC is to demonstrate how processes lead to common desirable outcomes. TofCs are useful for predicting how change is expected or conceptual tools for how the impact on care was achieved [34].

### Evaluation Design

Local VCSE partners were invited to take part in the study via the sponsor. Seven VCSE partners participated in the project, representing organisations from across the city. The VCSE partners provided a variety of services, including a drop-in cafe for people of African–Caribbean heritage, weight management programmes in groups and/or online, exercise referral programmes, diabetes education in the Urdu language, outreach activities for the Roma–Slovak community, men’s health groups and wellbeing clinics at local community centres. A co-ordinating voluntary services organisation commissioned the evaluation from an academic team. Their complex programme of work was ongoing and sought to identify the impact of VCSEs offering a health-oriented service to people with T2DM. The sponsor sought to co-produce a TofC from the existing work of VCSE organisations that participate in the delivery of T2DM services as part of the statutory care pathway. To be included, the VCSE had to offer a service in a specific underserved community and have pre-existing embedded connections and relationships with ethnic groups or socially or financially deprived areas. The evaluation team used the appreciative inquiry (AI) model as an evaluation framework because the process examines, identifies and further develops the best of what is happening [35]. This enables actors to co-create a shared understanding of, in this case, how VCSE organisations could work together productively on behalf of those with T2DM in a northern city in England. The evaluation design was co-produced [36] with the participating VCSE organisations familiar with the communities of interest but less experienced in research processes. Therefore, the interview methods and data collection were fully discussed and ‘rehearsed’ with the participant organisations. 

## 2. Materials and Methods

The theoretical approach to the study was based on organisational development and systems thinking used in health and care services in the UK [37]. This recognises that the individual is the central focus, but that healthcare is made up of multiple organisations and this broad context was important for this study.

The AI approach provides a framework for data collection based on the 4D cycle, i.e., discovery (what is the best of what it is), dream (envision the impact), design (co-production) and destiny (how to empower, learn and improvise). The participation of the seven VCSE services was particularly welcomed, in part to engage and build capacity for evaluation practices; in AI terms building “images of possibility into reality and belief into practice” [38]. In a community context, the method is aligned with an asset-based approach [39] where atypical populations are engaged and more marginal views are sought [40]. The evaluation had a pragmatic outcome of preparing the VCSE organisations for a joint application for further funding. In order to position organisations to apply for more strategic funding, the evaluation of practices also sought to generate a shared understanding of how VCSE outcomes could be defined and what benefits might be measured in relation to their practices [34]. The approach enabled organisations to share feedback from their constituent membership, and this data were analysed for themes and subsequently re-reviewed for the purpose of collaborating on the TofC development.

The research team comprised a dietitian (LN), an occupational therapist and systems researcher (SFD), a community and practice nurse (AY), a director of voluntary services organisation (HS) and a senior research fellow in global public health (SB).

### 2.1. Data Collection

A semi-structured interview guide was co-produced and designed in collaboration with the VCSE partners to ensure appropriate questions and prompts were used to gather the required information. It was developed to lead participants from their initial diagnosis to the present day, with specific questions regarding what having diabetes meant to them, how they manage their diabetes, where they access support and appropriate advice, what they liked and what could be improved regarding their experiences with primary care, secondary care and the VCSE sector regarding their diabetes. They also were asked what they felt would be important when designing a diabetes service for their community. The full interview guide is provided in the Appendix A. 

VCSE organisations recruited participants (either face-to-face or via telephone or e-mail) if they had received support and advice from the VCSE and had a diagnosis of T2DM. Participants were recruited and interviewed between December 2021 and February 2022 when COVID-19 rules were variable in the UK. Participants were given a detailed participant information sheet before the study commencement, with information about the voluntary nature of the input, assurance of data anonymisation and maintenance of confidentiality. Consent was confirmed before participation in the evaluation. Participants were free to withdraw from the study at any point during the process without impacting their routine care.

Demographic questionnaires and qualitative semi-structured individual interviews were used to collect data for this study. VCSE partners and staff members conducted the interviews. The project team members (LN and SB) provided training to the interviewers before they interviewed the participants. All interviewers completed a pilot questionnaire, and the project team provided feedback on their interviewing process and skills. The interviewers then completed their questionnaires and qualitative interviews with two to seven participants recruited by the VCSE organisations. The influence of the interviewers on the research processes and data could not be assessed as they needed to maintain field diaries or document their observations to achieve this.

The questionnaires were designed to collate basic demographic information such as age, gender, postcode, time since diagnosis and education status.

The interviews were conducted online, face-to-face or via telephone using a predetermined co-developed topic guide; each lasted between 20 and 40 min. After obtaining informed consent, the interviews were audio-recorded or hand-written verbatim, with some interviews translated from Urdu to English. The research team (SFD, LN and SB) received anonymised voice files which were transcribed by Otter.ai transcription software [41]. The research team immersed themselves in the data by listening to the audio interviews and reading and editing the transcripts produced by Otter.ai. 

### 2.2. Data Analysis

The data were analysed by the research team using the qualitative framework analysis approach [42,43,44]. This approach was developed in an applied research context to systematically manage qualitative data to identify the potential for actionable outcomes by providing transparent results and conclusions that can be related to the original data.

Three researchers (SFD, LN and SB) analysed the data manually by following the five steps in framework analysis: (a) data familiarisation: the researchers read the transcripts several times to develop an understanding and interpretation of the participant’s perceptions of their diabetes journey; (b) framework identification: each researcher suggested a theme heading for the framework analysis and these were discussed and amended until a consensus was reached; (c) indexing: condensed data from the transcripts; (d) ‘charting’; and (e) mapping and interpretation [43,44]. Following Gale et al. (2013), quotes were extracted from the transcripts to populate the framework. The framework interpretation involved reviewing and discussing the data and developing a TofC [43]. The data are reported using Standards for Reporting Qualitative Research (SRQR) using the Consolidated Criteria for Reporting Qualitative studies (COREQ) based on a 32-item checklist [45].

### 2.3. The Theory of Change

The initial TofC was developed from the framework analysis, with the themes forming the basis of further discussion between the research team, the participating VCSE organisations and the coordinating VCSE agency. The charted data and thematic descriptors were shared and used to highlight the collated impact of VCSE interventions [46]; the discussion described the process and the change sequence to consolidate the outcomes and impact on the T2DM population. Using ‘sense-checking’ through participant validation [47] of the framework analysis enabled a deeper understanding of the context of the intervention, particularly the cultural diversity of both the population and VCSE focus. The results were also presented to the wider VCSE community and National Health Service (NHS) organisations working to improve the management of T2DM in the area. A robust TofC model can be evaluated in subsequent studies. Still, the TofC provided a preliminary illustration by clearly mapping out the process between the intervention and desired outcomes [48]. 

### 2.4. Patient and Public Involvement

Participant organisations working within local communities were invited to sessions to scope the research area, agree on the key research topics and devise, pilot and agree on the interview topic guide. The local VCSE organisations’ interviewers recruited community members to understand their lived experience of T2DM in the local setting. 

## 3. Results

### 3.1. The Study Participants

In total, 33 interviews were collated from seven organisations and 30 were included in the analysis (2 interviews were excluded as the interviewees had type 1 diabetes and 1 interview was a duplicate). The sociodemographic characteristics of the participants are described in Table 1.

Participants were diagnosed between 12 months and over 30 years ago, with most participants being diagnosed within the last 5 years. Nearly half of the participants lived in the top 20% most deprived centiles, according to the English Indices of Multiple Deprivation (2019) from postcode data.

### 3.2. Identified Themes

Six themes were identified overall: individual and group support for self-management; trusted services and relationships across the community; long-term engagement with services; sociocultural context of diet and nutritional choices; multifaceted adaptation to the long-term condition; and shared community support network.

#### 3.2.1. Individual and Group Support for Self-Management

The study participants reported several ways they were given initial information, which they thought was generic but a good start. Most said that they commonly received early advice and guidance, but the overall messaging was associated with nutrition and lifestyle. For example, a White British male participant (65+ years old) said, 

“they told me to watch my diet, watch what I eat-plenty greens, fewer carbs, and to exercise, and that will keep my blood sugars down”.

However, many of the interviewees mentioned that they preferred tailored or specific information that could meet their individual needs.

In the initial information, the participants were offered medication such as Metformin and referred to DESMOND. DESMOND was attended and received variably by the participants. Due to ongoing comorbidities, there were many barriers to attending DESMOND, such as timings, location and inaccessibility. As a participant expressed his concern that: 

“I was invited to some talks, but I didn’t go. I was overweight and struggled to walk, and nobody could take me in their car” (Asian British Male, 65+ years).

However, some that did attend found it useful and reiterated their existing knowledge. For example, a Black Caribbean female (age 65+ years) mentioned that: 

“I was signposted to DESMOND. The programme affirmed things I know. It was useful”

Therefore, when attending the programme, she found it to be a good source of information for diabetes self-management. 

Nevertheless, some participants felt the information was ad hoc and only sometimes available when needed. Similarly, many respondents felt overwhelmed by the information they were provided with at diagnosis and could not initiate any positive behaviour change. When the shock of the diagnosis had settled, and they could act on the information, they felt that additional support was lacking. A participant expressed his concern as:

“…very disjointed, and it’s difficult to find out about that. I tended only to find out about things by chance, in some instances through desperation” (White British Male, 65+ years).

It was evident from the interview data that there was shared knowledge in families and across communities. In the absence of other information, this was also used to supplement understanding from health professionals. A Pakistani female (65+ years) said: 

“My GP told me to lose weight and stop eating rice and chapati and other things. My sister also had diabetes a few years before, and she also told me some tips”.

However, following initial advice from the GP or the DESMOND programme, many participants stated that there was confusion and uncertainty associated with the long-term effects of living with diabetes and what strategies to take forward in daily living. The advice was limited and from mixed quality sources. As a participant shared:

“At present, I don’t get any [information]. I am just living with the original information I was given about diet and taking one tablet until today. It concerns me because I don’t know whether my blood sugar is gone” (White British Male, 65+ years).

For those who could use the internet for information about the management of their diabetes, the quality of the information was varied, and some of the behaviour changes they took from it were inconsistent with usual dietary advice. A White British male (aged 65+ years) found it useful and emphasised that, 

“I think it has to be the internet, to be honest. Yeah. I found a lot on the internet about various foods and how they affect your blood sugars”.

Many participants reported finding answers and support from neighbours, friends, the local community, and voluntary organisations. This support was tailored to the community and individuals’ needs and delivered when the participants were ready to accept the information to improve their diabetes knowledge or control. A Black Caribbean female (age 65+ years) participant said that:

“And it also happened that she [neighbour] had diabetes. And she says to me, “oh, there’s a group going on at [name of VCSE]”… And I felt renewed, elated because it was a group that, that there were other people with the same diagnosis, the same struggle”.

Most participants stated that they received formal and informal information from various sources such as healthcare professionals, DESMOND, peers, relatives, family members and the internet. Most of the participants valued the information they received from their family, community or peers as they believed it was based on the lived experience of people, and they found it relevant to their lived experience. 

#### 3.2.2. Trusted Services and Relationships across the Community

There was a strong recognition by the participants of “good” services that they trusted and those they felt had let them down. Many of the services also seemed to have been impacted by the COVID pandemic, so the patients felt they were receiving a substandard service. A participant said:

“Diabetic nurse.... Well, it’s just generally sent me for bloods, look at my feet. Quick chat. Jobs a good ‘un!” (Unknown).

In contrast, a White British male (age 65+ years) felt that: 

“at the moment, due to the pandemic, no one [is supporting with diabetes management]; that’s what I feel let down by because I think if you’ve got something like diabetes, I should be getting support”.

Characteristics of the health professionals and services which participants appreciated were regular contact, building a good rapport, being personalised/culturally appropriate and referring or signposting them to services in the local community which were delivered primarily by the VCSE organisations. For example, an Asian British female (56–65 years) stated that: 

“the information I was given was appropriate for me. I visited the nurse at the doctor’s surgery every 3 months for my bloods, she was very good, and I developed a good relationship with her. She told me about exercise classes in the area and also told me about health walks”. 

Some participants also discussed how their personal circumstances prevented them from attending some of the services or how they avoided appointments because they knew that they had poorly controlled diabetes. As a mixed heritage male (26–35 years) said: 

“my mental health is really bad due to not managing diabetes. It has affected my life. I try to avoid my GP appointments because I know I struggle to manage my diabetes”. 

Many respondents also spoke about the VCSE organisations they had received support from and how it improved their knowledge of diabetes control. As a participant said, 

“I get most of my information now from community hub, I go for the keep fit sessions, and they do some health information cafes where they talk about healthy things to eat and portion sizes. They also tell you what food to avoid if you have diabetes. It has been a good refresher for me”. (Pakistani Female, 65+ years).

The level of participants’ trust in service providers and healthcare professionals varied. Still, most patients trusted those services and healthcare professionals who listened to them and addressed their individual needs. 

#### 3.2.3. Long-Term Engagement with Services

The diabetes journey reported by many of the participants was similar. They had gone to the GP with symptoms of thirst, tiredness, poor wound healing, etc., and were diagnosed with diabetes. For many, this was expected, and participants reported having parents and siblings with diabetes. Participants linked their diabetes journey to their family history and said, 

“I had a couple of things that I spoke to a doctor about: dehydration, night sweat, that kind of thing. And they did a blood test. So, it was a blood test that revealed. Like yeah, because yeah, my mom’s diabetic. And then I’ve always known it was coming. Yeah, it was just a case of, obviously, two brothers. And I was never sure if it was going to be me or them too. But it seems to have come down the female line” (White British Female, 56–64 years).

However, for some participants, this was a complete shock due to their perceived idea of who gets diabetes, and this caused a large biographical disruption. For instance, for an Asian British female (56–65 years), it was a shock when she learnt that she had diabetes. She shared that: 

“the doctor sent me for blood tests, and they came back showing I had diabetes. I was so shocked and upset. I didn’t even know young people got diabetes”. 

Most participants expressed that the diabetes diagnoses often led to strong emotional responses where people were stressed and upset. It took a long time for them to process their new diagnosis and accept what this might mean to them. As it was for a Pakistan female (65+), 

“I was very, very upset when I found out I had diabetes. I was scared that I would not be able to eat all the food that I loved and worried about what I would eat”.

After the initial support and advice that was offered, participants reported a period of struggling, anxiety and uncertainty. The reported impact of the diagnosis on their mental health was incredibly significant. It often prevented them from processing or using the diabetes management information they were provided with at this time to make any meaningful lifestyle change. An Asian British female (56–65 years) shared her experience, 

“The first few years were very tough. I stopped eating out and I stopped visiting parties and friends because I was worried that they would offer me food and I wouldn’t be able to eat anything. I was also scared that people would be watching me and watching what I eat. This made me isolated and depressed, and I developed social anxiety. It has taken me years to get over this, and now I feel I am confident and in control”.

The participants also reported ongoing concerns about the long-term implications and co-morbidities associated with their diabetes diagnosis. For example, 

“I didn’t really take on board or understand how serious diabetes is. I thought if I take my tablets, watch my diet, and look after my feet, I’d be ok…. but talking to people, they tell me about problems with their kidneys and blood flow to their legs, lots of problems. I hope I don’t get any of this” (Black Caribbean British Female, 65+ years).

The participants had mixed feelings and responses when they were diagnosed with diabetes. Where participants had a family history of diabetes, they knew that they would have diabetes too; however, some of them were shocked when they were diagnosed with it and had a need for long-term support. 

#### 3.2.4. Sociocultural Context of Diet and Nutritional Choices

The interview data revealed that some participants were self-motivated or had household support that seemed to be a significant beneficial factor in their response to the diagnosis by driving more appropriate food choices and aiding weight loss, whilst others relied on food banks and struggled to make appropriate food choices due to the limited availability of foods. As an Asian British Female (56–64) said, 

“My husband was very helpful and supportive...he helped me reduce some weight…I realised that when I cut down on sugary stuff and started eating less rice and chapati, my weight dropped by itself. I also started walking a lot, and me and my husband would go for walks after the evening meal.”

Moreover, the participants spoke about dietary changes they had been advised about or initiated. It was apparent that carbohydrate-rich foods are ones that people regard as bad for them, and the participants have engaged with the idea of eating more fruit and vegetables. However, the messaging they had retained from the advice appeared generic and only sometimes helped the individuals make a specific dietary change suitable for them. As a White British male (56–64 years) said, 

“Anyway, apparently, you’re not supposed to have carbohydrates and sugars, which is, like, most of what I eat, potatoes, bread”.

In addition, advice from healthcare professionals was combined with advice from other sources such as the internet, family and friends. For instance, a White British male (65+ years) stated that:

“I eat a lot of onions and green onions and leeks and that kind of thing…very particular triggers and learning how to regulate how much fruit I eat, shouldn’t eat too much fruit, certain fruits you should avoid, and so on. Yeah, I learned a lot from the internet”.

It was evident from the interviews that most participants were conscious of their diet and followed the nutritional advice they received. Moreover, they mentioned a variety of sources of information concerning diet, including healthcare professionals, family, friends and neighbours, but were only able to enact behaviour change within their social, economic and cultural resource constraints. 

#### 3.2.5. Multifaceted Adaptation to the Long-Term Condition

Most participants were users of VCSE services and appreciated the focus on local knowledge and shared information in a familiar setting. The support was focused on practical daily activities and was derived from discussing and planning with others with similar concerns. For example, a Black British Caribbean female (65+ years) said: 

“The information I got was from …. a really good group. You know, it’s, it helps me really, to manage.. day -to-day menu, day-to--to-day exercising, everything. I mean, every…everyone that was there was going through the same thing. I mean, the shared experience, we all shared our experience, you know, because everybody was different there. And they understood what everybody else had going on. So, it helped”.

A common consideration was that participants had multiple morbidities and often multiple stresses arising from household challenges, including social stress and deprivation. These competing priorities made lifestyle behaviour change difficult for participants, resulting in a deprioritisation of diabetes self-management. For some, the initial stages following diagnosis were easier to manage, involving weight loss and a degree of support via the DESMOND programme. Still, the progress of the disease was difficult to manage, and there were varying responses—some more positive than others—but predominantly associated with medication changes or blood checks. A participant said:

“I immediately started walking and eating more fruits and vegetables and lost a lot of weight. I controlled it via diet for at least 2 years. After 2 years, I was put on medication. I am still learning about things even at this stage. I have stopped eating all sugary stuff; I have a lot of self-control” (Pakistani Female, 65+ years).

Home remedies were recognised as alternatives to allopathic medicine and a part of active self-management, and some participants favoured mixing these with their medications and lifestyle choices. For example, a Pakistani female (65+ years) shared examples of using home remedies: 

“I also tried some bitter drinks that people in Pakistan drink when they have diabetes, that helps me keep my sugar level under control, and I also eat bitter gourds, that is a very bitter vegetable. I also put cinnamon in my tea as I have heard that it helps lower when levels are high. I am very careful I don’t let my levels go high”.

In an extreme case, the self-management of diabetes was complicated by other habits and problems or a sense that there were limited options concerning being well and managing their lifestyle. One man reported having a perforated eardrum. 

“So, there’s nowt [nothing] he’s going to do for me, says you…you’ve got diabetes, and you smoke. So anyway, so basically, I’m just left here with a perforated eardrum,” said a White British Male (56–64 years).

As a part of active self-management, there was a range of comments about the continuity of activities, what participants liked to do, and what they stopped doing—particularly associated with exercise. In many cases, people wanted to move but were out of practice, had their routine altered due to COVID, found it difficult to find time or were too embarrassed to use municipal facilities.

#### 3.2.6. Shared Community Support Network 

Following the appreciative inquiry model, the participants were asked how they envision existing services’ improvements and impacts. The participants from black and ethnic minority backgrounds dreamt of being recognised as having a cultural heritage and a health support need. For example, older adults from Pakistan wanted information in Urdu, which they saw as the only culturally appropriate way to overcome language barriers. On the other hand, there was recognition that some groups needed to include collective knowledge about health. As an Asian British male (46–55 years) put it, 

“There is a need for a lot of improvement culturally because then if we introduce…traditionally if there is one person gets the information, he spread it out to the [the rest of the community]… a big job to for the people’s awareness in mainly an Asian community”.

Both participants from ethnic minority and white backgrounds suggested they would like simple but meaningful practical formats for health advice, including regular exercise and cooking support, that were associated with a healthy lifestyle but delivered in ways specific to the community. For instance, a Black British Caribbean woman (65+ years) wanted to incorporate exercise, with different topics and ideas about how she would take the group outside to get fresh air. 

The social and collective stimulus was repeated with the suggestion that groups learn, support and share informal information in a way that promotes long-term communication associated with changing personal behaviours. A participant emphasised that

“we need to meet up regularly and consistently. Peer support and good communication” (Black British Caribbean Male, 56–64 years).

The social nature of these interventions was associated with inclusion and, for some, was not exclusively about their diabetes but was a ‘place-to-be’, where a sense of belonging was experienced. For example, a White British man (46–55 years) reflected that: 

“A safe place, a comfortable place…. I’d like to make a place where anybody can come. You don’t have to talk about it if you just want to come to sit. Have a cup of coffee—a bit like the men’s group”.

Many wanted to see more proactive services associated with primary care. These were associated with the need for physical checks and professional advice from a service based on technical knowledge. There was a suggestion that the technical knowledge must be complemented with a different communication style and a more nuanced, local and community understanding. Overall, there was consensus among the participants concerning using and sharing collective wisdom and experiences to have culturally appropriate and accessible spaces that can stimulate participants to embark on active management of T2DM. 

## 4. Discussion

This evaluation used AI and qualitative framework analysis synthesised into a TofC ([33,48]) to provide an explanatory process for the impact of VCSE services that support people with T2DM in underserved communities. A TofC diagram was produced with the participants and the sponsor to evidence the impact of the combined VCSE services. The TofC was based on themes and theme content that reflected the current status of the service offer, the short- and long-term outcomes which were elicited from the conversations and interviews and the impact which was derived from the interview participants’ data. The TofC restates the strategic aim that VCSE organisations aim to operate in collaboration across the health and care sector and economy, offering supportive guidance to diverse communities to support and improve self-management of T2DM (see Figure 1).

The TofC proposes that trusted local VCSE services are embedded into the diabetes care pathway alongside their primary and secondary care partners. The VCSE’s appear to offer nuanced information that can be delivered at a time and in a way that meets the needs of underserved communities and thus may be a more effective model of working. Short-term outcomes involve the use of current services by marginal communities, whilst longer-term service development and delivery may be undertaken by individuals from such communities, representing a deeper understanding of the needs of the population. It is noted that the multicultural service models and staffing are aligned to the community, and the sense of belonging afforded to communities by VCSE organisations is significant to the perceived impact, with VCSEs acting as coordinators across the community to achieve sustainable services which work for underserved communities within a complex health system.

The themes contributing to the TofC suggest that a combination of statutory and VCSE services is available, but that many participants in this evaluation preferred to use ‘local’ advice and more community ‘embedded’ knowledge. An important finding was that the needs of individuals changed whilst living with diabetes. Initially, people with T2DM seek reassurance at an early stage post-diagnosis; however, latterly, they need to ‘check in’ and appreciate ‘technical’ support to monitor and manage the disease process. Primary care is recognised as the professional service and emphasises patient-centred, comprehensive care where communities are underserved, as is often the case with marginal and diverse ethnic groups [21]. However, the assets within the local communities make up for some of the perceived lack of personalisation and support from statutory services. 

The findings of the study are reflective of previous research; the Care Quality Commission (CQC) report ‘My diabetes, my care’ (2016) recommended the development of local plans to put people at the centre of their own care, with appropriate levels of support for self-management and ensuring emotional and cultural considerations are met [49]. Similarly, the NHS Confederation (2021) recommends that “Local VCSE organisations need to be included in health and care pathways and service redesign planning across systems, including population health management and social prescribing in primary care networks” [2]. Debussche et al. (2022) also recommended that interventions and services designed for people living with T2DM need to consider the multidimensional nature of diabetes self-management and health literacy as well as the social context of the individuals for it to deliver effective outcomes [50]. By giving a voice to underserved communities with lived experience in service design and delivery, systems can be more effective, reduce health inequalities and deliver meaningful outcomes [51].

VCSE services receive marginal and short-term funding to deliver pockets of activity [7]. Their role is often one of an advocate, enabling communities to meet, share experiences and voice their concerns in a socially constructed space [1]. The assets are created, sometimes in the short term, co-produced with the community and so often meet a need that a healthcare provider cannot. However, the provision is weak by virtue of its organisational form: small, responsive and contractually regulated. This project usefully highlights the value of the core purpose: engagement and association with marginalised communities and management of the associated risk. It also highlights that the VCSE services are bridging the gap between primary care and population needs. However, the input from the VCSE needs to be more cohesive and consistently meet the needs of its community, which is challenging due to limited resources, infrastructure and expertise. As the ICSs integrate traditional health and care systems with the VCSE sector, clinicians may be more closely aligned with community assets and be able to signpost patients to additional support for diabetes self-management which may ensure more tailored, targeted and culturally competent practice and improved health outcomes for those most marginalized communities.

The strengths of this study lie in the collaborative working and co-design of the research tool and protocol between different organisations, which has built on existing relationships with VCSE providers and enabled access to seldom heard and more marginalised patient groups. The findings and the TofC were also shared with all VCSE groups and stakeholders within the region to obtain feedback and ensure that the process was transparent and representative of the data collection. If a strong collaborative process is established in the initial stages of a TofC evaluation, it can help assure that stakeholders will work with the evaluator to explore creative measurement strategies [52]. However, additional tools are required which will help to scale up successful coproduction interventions for new people within new contexts or place. There is also a challenge to provide appropriate development and professional practice skills and knowledge to current and future professionals in order for them to engage and confidently work with coproduction approaches [53].

The study methodology had some limitations, as the participants were interviewed by staff members from a VCSE service that they had accessed, which may have constrained what they said about the organisation. There may have been selection bias towards those individuals who experienced a VCSE service more positively, and these individuals were likely to be more engaged with their own self-management of diabetes as they were accessing healthcare and VCSE organisations. However, the researchers believe this elicited high-quality data due to the prior relationship with the ‘trusted’ organisation and the interviews were not generally carried out by the participants’ usual contact at the VCSE organisation. Using non-academic researchers for data collection led to some inconsistencies in data collection. Still, the benefits outweigh the limitations, and the interview data collected are equally critical for all diabetes care providers. Other limitations may relate to the specific context and configuration of VCSE services that may not be replicated in other countries. Whilst the findings are recognised to be context specific, and have no clinical implications, they do have implications for changes in health and care utilization. Additional research to understand the perceptions of those who have not engaged with VCSE services would also be beneficial to determine factors which lead to engagement with such services. The initial TofC seeks to map the impact of services, and the study demonstrates again the preferences for tailored person-centred interventions delivered in a local context. There is value in assessing whether this TofC is replicable in other contexts, including different cultures and other long-term health conditions, and in assessing the impact of the VCSE sector on reducing the NHS burden.

## 5. Conclusions

Participants with T2DM presented challenges associated with their diagnosis and early management and appreciated the consistent and embedded community support of the VCSE providers. Participants reported an inconsistent range of support post-T2DM diagnosis and very limited engagement with the main educational programme offered as a standard (e.g., DESMOND). The evaluation identified the benefits of hyper-local group activities, often employing peer-to-peer processes to share information about living with T2DM and providing support at a time which suited them. Advice, guidance and long-term support were only sometimes available but deemed important. The strength of the impact of VCSE organisations was associated with the engagement and continuous relationships with individuals, knowledge of the community and culturally appropriate provision (concerning language and cultural diets); there was still a demand for more responsive and cohesive support for lifestyle and behaviour changes. The services appear to strengthen the specific lifestyle behaviour change based on ‘what mattered’ to individuals—recognising and overcoming some barriers to accessing statutory care. 

The findings highlight a systemic and organisational issue which can be resolved by a cross-sector integrated health and care system which embeds the VCSE organisations into the diabetes care pathway. The TofC indicates that with consistent, sustainable funding and recognition, VCSE organisations can contribute valuable services targeted to more marginal and underserved communities. Other competencies through health coaching training or similar may be helpful and require collaboration across the sector to enable a specific response to diabetes management.

## Figures and Tables

**Figure 1 healthcare-11-02499-f001:**
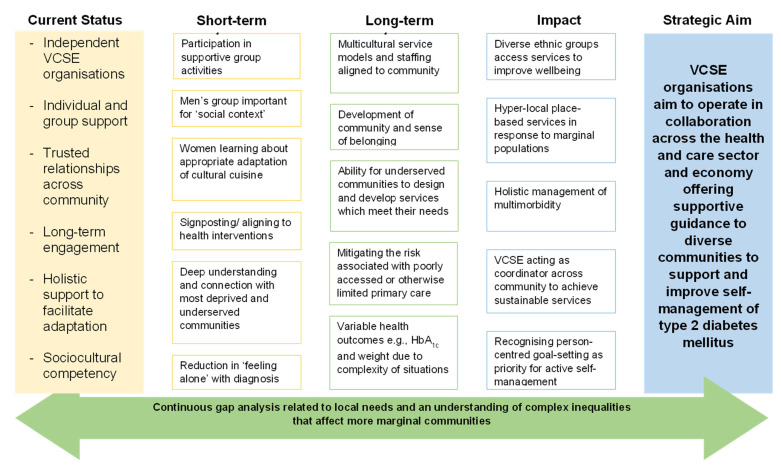
Initial theory of change for VCSE organisations in diabetes management.

**Table 1 healthcare-11-02499-t001:** Participant characteristics and demographic data (*n* = 30).

Sex (Male/Female) (%)	14/16 (47%/53%)	
Age (years) (%)	26–45	4 (13%)
	46–55	8 (27%)
	56–65	5 (17%)
	65+	9 (30%)
	Unknown	4 (13%)
Ethnicity (%)	Asian or Asian British	4 (13%)
	Black or Black British	7 (23%)
	Mixed—Other	3 (10%)
	White—British	11 (37%)
	White—Other	1 (3%)
	Unknown	4 (13%)
Education status (%)	No formal qualifications	2 (7%)
	Up to GCSE or equivalent	5 (17%)
	AS/A level or equivalent	2 (7%)
	Apprenticeship	1 (3%)
	Further Education	6 (20%)
	Undergraduate degree	4 (13%)
	Postgraduate degree	1 (3%)
	Prefer not to say/unanswered	9 (30%)
Time since diagnosis (years) (%)	0–4.9	13 (43%)
	5–9.9	2 (7%)
	10–14.9	5 (17%)
	15–19.9	1 (3%)
	20+	2 (7%)
	Prefer not to say/unanswered	7 (23%)
English Indices of Multiple Deprivation (2019) [by postcode data] (%)	0–20% most deprived	14 (47%)
	21–50%	7 (23%)
	51–80%	1 (3%)
	20% least deprived	2 (7%)
	Prefer not to say/unanswered	6 (20%)

## Data Availability

All data created during this research are openly available from the Sheffield Hallam University Research Data Archive (SHUDRA) at http://doi.org/10.17032/shu-180038.

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
