# Peer review of "Impact of Voluntary, Community and Social Enterprise (VCSE) Organisations Working with Underserved Communities with Type 2 Diabetes Mellitus in England"

_healthcare, 2023, doi:10.3390/healthcare11182499_

Round 1
Reviewer 1 Report
This is a significant study that evaluated a self-management program for people with T2DM to identify best practice and develop a theory of change. It is unique in that the evaluation of a chronic disease prevention and management program in primary care was designed and implemented with the participation of community stakeholders. In addition, this evaluation study explores the role of VCSEs and suggests directions for embedding them in the primary care system, which adds to the value of this paper.
However, the paper could be enhanced with a few modifications.
Firstly, as this is an evaluation of a culturally appropriate diabetes management program of the NHS in England, the title, abstract, and keywords should include the case of "England."
Secondly, in order to improve the understanding of international readers, the introduction section should explain how the role of VCSEs is changing in the UK NHS system and how it compares to global trends.
Finally, when presenting the results of qualitative analysis, direct quotations should be formatted appropriately, including indentation, so that they can be separated from the narrative.
I advise you to correct typos when submitting revisions.
Author Response
Please see the response to reviewer 1 letter attached.

Reviewer 2 Report
This might be a useful paper, although the presentation of "findings" makes any final judgment difficult.
Firstly, I think that the phrase (after the colon in the title) should be "An English Case Study". The findings are unclear and uncertain, although they do give the impression that this provisional or pilot study might lead to useful further work.
Initials such as NICE refer to a UK institution, and these initials should be spelled out, as well as the general context of health care access and delivery in the UK. Might the findings have implications for health care management in other cultures?
Do the findings have any clinical implications (e.g. in terms of diet and medication for diabetes self-care)? What is the professional qualification of the authors of this study (eg in nursing, epidemiology, nutrition etc)? Do the findings have actual clinical implications for the management of T2DM in underserved populations? Did change in self-care and clinical care actually occur? But would not this require follow-up interviews to make an adequate assessment?
The theoretical basis for the study is not sufficiently elaborated, with only a few references, which are not properly elaborated, in the text. The methodology of conducting and analysing qualitative interviews is not adequately explained.
Reference to [38] Otter is incomplete and puzzling.
Reference [46] is obscure, and I was unable to find any link in published literature which would allow me to read it. What is CQC? Is it an earlier publication of these results?
My opinion is that this work should be reframed as a pilot study, with methodological directions for a larger, subsequent study.
Author Response
Please see our response to reviewer 2 letter attached.

Round 2
Reviewer 2 Report
The revisions made make this an acceptable article.